# Longan Flower Ethanol Extract, *Dimocarpus longan* Lour, Mitigates Oxidative Damage and Inflammatory Responses While Promoting Sleep-Related Enzymes in Cell Models

**DOI:** 10.3390/biomedicines13071588

**Published:** 2025-06-29

**Authors:** Chao-Chun Ma, Ming-Chang Hsieh, Wei-Lun Chiang, Yi-Wen Chen, Pin-Chao Huang, Chin-Hsiu Yu, Shao-Yu Lee, Tin-Ching Chung, Hsi-Chi Lu, Yu-Wei Chang

**Affiliations:** 1Department of Medicine Laboratory, Jen-Ai Hospital, Taichung 412224, Taiwan; chaochun1026@gmail.com; 2Department of Medical Laboratory and Biotechnology, Chung Shan Medical University, Taichung 402306, Taiwan; cshb183@csh.org.tw (M.-C.H.); miss9ch@gmail.com (W.-L.C.); rainbowchung.916@gmail.com (T.-C.C.); 3Department of Clinical Laboratory, Chung Shan Medical University Hospital, Taichung 402306, Taiwan; 4Department of Food Science, Tunghai University, Taichung 407224, Taiwan; yiwen1119@thu.edu.tw; 5Percheron Bioceutical Research Center, Taichung 407612, Taiwan; stanley.huang@nutrarex.com.tw (P.-C.H.); yuna.yu@nutrarex.com.tw (C.-H.Y.); shaoyu.lee@nutrarex.com.tw (S.-Y.L.)

**Keywords:** sleep disorder, longan flower extract, inflammation, antioxidant

## Abstract

**Objectives:** Modern lifestyles factors such as digital overload, aging, and poor sleep hygiene have led to increasing cases of sleep disturbances and inflammation-related disorders. These conditions are often associated with oxidative stress and immune dysregulation. Longan flower extract (LFE), traditionally used in East Asian medicine, has shown potential health benefits but remains scientifically underexplored. This study aims to investigate the chemical composition and bioactive effects of LFE on inflammation, oxidative stress, and melatonin biosynthesis in relevant cellular models. **Methods:** LFE was prepared using ethanol extraction and characterized for its total polyphenols, flavonoids, oligomeric proanthocyanidins (OPCs), and corilagin content via HPLC. Its anti-inflammatory, antioxidant, and neuroregulatory activities were assessed in LPS-stimulated RAW 264.7 macrophages and BV-2 microglial cells. Key assays included quantification of cytokines (TNF-α, IL-6), detection of nitric oxide (NO) and reactive oxygen species (ROS), and measurement of antioxidant enzyme activities (GPx, SOD). Gene expression of melatonin biosynthesis enzymes was evaluated using quantitative PCR. **Results:** LFE treatment significantly reduced LPS-induced TNF-α, IL-6, NO, and ROS production in both cell models. It enhanced GPx and SOD activity and increased intracellular glutathione levels. Moreover, LFE upregulated the expression of *TPH1*, *DDC*, *AANAT*, and *ASMT*, genes involved in melatonin biosynthesis, and promoted serotonin secretion. **Conclusions:** These findings suggest that LFE holds significant potential as a natural therapeutic supplement, particularly for alleviating sleep disturbances, reducing oxidative stress, and modulating inflammatory responses associated with modern lifestyle-related health conditions.

## 1. Introduction

Contemporary society’s rapid pace imposes substantial psychological and physiological stress on human health. Recent epidemiological data demonstrate a concerning trend: the prevalence of insufficient sleep (defined as less than six hours per night) is increasing significantly among adults, primarily attributed to excessive digital device usage and suboptimal lifestyle habits [1]. Of particular concern, this phenomenon has extended to younger demographics, with both adolescents and children exhibiting comparable sleep disturbances [2]. Current statistical analyses indicate that approximately 30–50% of adults in the United States experience insomnia symptoms [3,4]. Furthermore, research has established that endogenous melatonin production demonstrates an age-dependent decline [5], which may significantly impact sleep regulation. This physiological decline, combined with age-related life stressors, creates a vicious cycle where poor sleep disrupts cognitive function, metabolism, and immune response, which, in turn, leads to even worse sleep quality.

Sleep disorders like insomnia are established risk factors for metabolic diseases, including coronary heart disease, obesity, diabetes, and hypertension [6,7,8]. Poor sleep also strongly correlates with mental health conditions such as depression and anxiety [9,10]. Animal studies have shown that sleep deprivation increases oxidative stress and induces systemic and neuroinflammatory responses [11]. While people commonly turn to alcohol or melatonin supplements to manage sleep problems, alcohol is detrimental to health, and the long-term safety of melatonin, particularly in vulnerable populations, remains understudied [12]. These concerns highlight the growing need for non-pharmacological approaches to address modern lifestyle-related health challenges, particularly those involving sleep disturbances, chronic inflammation, and oxidative stress.

Fruits, vegetables, and edible flowers have been proven as safe, effective, and sustainable sources of natural antioxidants and free radical scavengers [13,14]. Active compounds found in fruits, vegetables, and edible flowers, such as phenolic acids, flavonoids, and anthocyanins, are widely recognized for their anti-inflammatory and antioxidant effects, which help to lower the risk of developing metabolic diseases [15]. Research shows that people generally trust natural food-derived health benefits more than synthetic drugs [16]. Since strict dietary and exercise regimens often face compliance challenges, developing scientifically validated natural food supplements into accessible nutraceuticals offers a practical approach to improving public health.

*Dimocarpus longan* Lour. (longan) is a renowned fruit native to East Asia and Australia. In traditional medicine, longan fruit pulp has been used to treat insomnia, promote mental calmness, relieve neurasthenia, and prevent cognitive decline [17]. Previous studies have primarily focused on longan fruit pulp’s bioactivities, demonstrating strong immunomodulatory and anticancer properties [17,18,19,20]. However, emerging studies also highlight the biological potential of longan flower extract (LFE), reporting antioxidant, anti-inflammatory, anticancer, anti-obesity, and hypolipidemic effects [21,22,23,24]. Despite these findings, longan flowers are still largely considered agricultural waste, and comprehensive studies characterizing their chemical composition and biological mechanisms remain limited.

This study aims to provide a comprehensive analysis of LFE, integrating phytochemical profiling with biological activity evaluation in immune and neuronal cell models. Specifically, we assess LFE’s modulatory effects on oxidative stress, inflammatory responses, and melatonin-related metabolic pathways, which have not been thoroughly explored in prior LFE studies. Our goal is to evaluate whether LFE could be developed into a functional nutraceutical to promote health, particularly by alleviating oxidative stress, modulating inflammation, and enhancing sleep-related biochemical pathways. While this study focuses on in vitro efficacy, future investigations will be needed to assess the safety profile and potential toxicological effects of LFE in vivo.

## 2. Materials and Methods

### 2.1. Preparation of Longan Flower Extract (LFE)

Longan (*Dimocarpus longan* Lour.) flowers were collected in Taichung, Taiwan, during full bloom in April 2023 from pesticide-free farms. Only mature, fully opened flowers were used. The fresh flowers were air-dried at 50 °C in the shade for 24 h and then pulverized into coarse powder. The dried flower powder was extracted with ethanol at a material-to-solvent ratio of 1:15 (*w*/*v*). Ethanol was selected because it is traditionally used in herbal medicine and is highly effective at extracting phenolic compounds and other bioactive constituents [25]. The mixture was heated at 65 °C for 60 min using a thermostatic water bath. The resulting extract was filtered through a 400-mesh filter, concentrated under reduced pressure using a rotary evaporator, and finally spray dried to obtain powdered LFE.

### 2.2. Analysis of Bioactive Compounds

Total polyphenol content was determined using the Folin–Ciocalteu method with minor modifications [26]. The spray-dried longan flower extract (LFE) powder was reconstituted in distilled water, and 0.1 mL of the sample was mixed with 0.5 mL of Folin–Ciocalteu’s reagent and 0.4 mL of 7.5% sodium carbonate solution. Following complete reaction, absorbance was measured at 750 nm.

Flavonoid content was quantified using a modified procedure [27], adapted to enhance sensitivity for low-concentration samples. Briefly, the sample (0.2 mL) was mixed with distilled water (2.4 mL) and 5% sodium nitrite (0.2 mL). After mixing, 10% aluminum nitrate (0.2 mL) was added. Once the reaction was complete, 4% NaOH (2 mL) was added, and absorbance was measured at 500 nm.

Oligomeric proanthocyanidins (OPCs) were determined by mixing the spray-dried LFE with 4% vanillin–methanol solution followed by concentrated hydrochloric acid under light-protected conditions [13]. After 15 min, the absorbance was measured at 500 nm.

All assays were performed using standard curves generated from their respective reference compounds—tannic acid for total polyphenols, quercetin for flavonoids, and catechin for proanthocyanidins—with correlation coefficients (R^2^) exceeding 0.95. The results were calculated based on absorbance values relative to these standard curves, following established phytochemical quantification protocols.

Gallic acid, corilagin, and ellagic acid were quantified by high-performance liquid chromatography (HPLC) using a Waters 2695 separation module coupled with a Waters 2996 photodiode array (PDA) detector (Waters Corp., Milford, MA, USA). Ten grams of dried longan flower was soaked in 200 mL of pure water, heated at 85 °C for 30 min, and filtered through a 0.45 μm membrane. This approach was used instead of the spray-dried LFE powder to preserve the integrity of heat-sensitive or volatile polyphenolic compounds that may degrade during spray drying. Using dried raw flowers directly provides a more accurate representation of the original phytochemical profile. After filtration, 20 μL of sample was injected into an Atlantis^®^ T3 HPLC column (5 μm, 4.6 × 250 mm) at 25 °C. Detection was conducted at 280 nm with a gradient elution program (Table 1) involving 0.1% formic acid in water (Phase A) and 0.1% formic acid in acetonitrile (Phase B) at a flow rate of 0.8 mL/min.

### 2.3. Cell Lines

RAW264.7 murine macrophage cells were obtained from the Bioresource Collection and Research Center (BCRC), Taiwan. SH-SY5Y human neuroblastoma cells were obtained from the American Type Culture Collection (ATCC, Manassas, VA, USA). BV-2 mouse microglial cells were provided by Taichung Veterans General Hospital (Taichung, Taiwan). QGP-1 human pancreatic cancer cells were obtained from the Japanese Collection of Research Bioresources (JCRB, Osaka, Japan). All cells were cultured according to the suppliers’ protocols.

### 2.4. Cell Viability Assay

Cell viability was assessed using AlamarBlue™ (Invitrogen, Carlsbad, CA, USA) [28]. Cells were seeded at 2.5 × 10^4^ cells per well in 96-well plates. After adhesion, the cells were washed with phosphate-buffered saline (PBS) and treated with LFE at concentrations of 0, 25, 50, 100, 200, 400, 800, and 1600 µg/mL for 24 h. Additionally, lipopolysaccharide (LPS)-only and medium-only groups were included as treatment controls. Subsequently, 10 μL of AlamarBlue reagent was added to each well and incubated at 37 °C. Absorbance was measured at 570 nm and 600 nm.

### 2.5. Measurement of TNF-α and IL-6

RAW264.7 and BV-2 cells were seeded at 2 × 10^5^ cells per well in 6-well plates, treated with LFE at concentrations of 12.5, 25, and 50 µg/mL for 24 h, and co-treated with 1 μg/mL LPS for an additional 24 h. Untreated cells served as the negative control, while cells treated with LPS alone (without LFE) served as the positive control. After treatment, the culture supernatants were collected and stored at −20 °C. Concentrations of TNF-α and IL-6 were measured using ELISA kits (Mouse IL-6 DuoSet ELISA and Mouse TNF-α DuoSet ELISA, R&D Systems, Minneapolis, MN, USA) according to the manufacturer’s instructions.

### 2.6. NO Assay

The concentration of nitric oxide (NO) was measured using the Nitrate/Nitrite Colorimetric Assay Kit (Cayman, Ann Arbor, MI, USA), following the manufacturer’s protocol. Briefly, cells were seeded into 6-well plates at a density of 2 × 10^5^ cells per well. After cell attachment, different concentrations of LFE were added, and the cells were incubated for 24 h. Subsequently, 1 μg/mL of LPS was added, and the cells were further incubated for an additional 24 h to induce inflammation. After treatment, 100 μL of the culture supernatant was collected and mixed sequentially with 50 μL of Griess Reagent R1 and 50 μL of Griess Reagent R2. The mixture was incubated at room temperature for 10 min in the dark. The absorbance was then measured at 540 nm using an ELISA reader. The NO concentration in the samples was calculated by referencing a standard curve generated from known concentrations of nitrate/nitrite standards.

### 2.7. Measurement of ROS

Intracellular reactive oxygen species (ROS) levels were evaluated using a commercial ROS Detection Cell-Based Assay Kit (DCFDA, Cayman, Ann Arbor, MI, USA), following the protocol provided by the manufacturer. Initially, cells were plated at a density of 1 × 10^4^ cells per well in 96-well plates and allowed to adhere overnight. Different concentrations of LFE were then administered for 24 h. Subsequently, the cells were challenged with 1 μg/mL LPS and incubated for an additional 24 h to induce oxidative stress. Post-treatment, the culture medium was aspirated, and the cells were gently rinsed with Cell-Based Assay Buffer to remove residual compounds. Thereafter, 100 μL of ROS Staining Buffer was added to each well, and the plates were incubated in the dark at 37 °C for 30 min to allow the staining reaction to proceed. Following incubation, the staining solution was discarded, and the cells were replenished with fresh assay buffer. Fluorescence intensity, indicative of intracellular ROS accumulation, was measured using a microplate reader with excitation at 480–500 nm and emission at 510–550 nm.

### 2.8. GPx Activity Assay

The activity of glutathione peroxidase (GPx) was determined using a commercially available Glutathione Peroxidase Assay Kit (Cayman Chemical, Ann Arbor, MI, USA), following the manufacturer’s instructions with minor adjustments. Cells were seeded into culture dishes at a density of 2 × 10^5^ cells per dish and treated with various concentrations of LFE for 24 h. Subsequently, the cells were stimulated with 1 μg/mL LPS for another 24 h to induce oxidative stress conditions. Thereafter, cells were detached using a sterile cell scraper and collected into tubes. The cell suspensions were centrifuged at 1500× *g* for 10 min at 4 °C. After carefully discarding the supernatant, the resulting cell pellets were resuspended in 100 μL of cold lysis buffer containing 50 mM Tris-HCl (pH 7.5), 5 mM EDTA, and 1 mM dithiothreitol (DTT). The suspensions were homogenized thoroughly and subjected to a second centrifugation at 10,000× *g* for 15 min at 4 °C. The resulting supernatants were collected and stored at −80 °C. For the GPx activity assay, 50 μL of assay buffer (50 mM Tris-HCl, pH 7.6, containing 5 mM EDTA), 50 μL of co-substrate mixture, and 50 μL of NADPH were sequentially added into each well of a 96-well microplate. Subsequently, 20 μL of either the sample supernatant or the standard solution was added to initiate the enzymatic reaction by the addition of 20 μL of cumene hydroperoxide. The decrease in absorbance at 340 nm, corresponding to the consumption of NADPH, was monitored kinetically at one-minute intervals using a microplate reader. GPx activity was calculated based on the rate of absorbance change over time.

### 2.9. SOD Activity Assay

The enzymatic activity of superoxide dismutase (SOD) was assessed utilizing a commercial Superoxide Dismutase Assay Kit (Cayman Chemical, Ann Arbor, MI, USA), following the manufacturer’s recommended protocol. Cells were seeded at a density of 2 × 10^5^ cells per dish and subjected to treatment with LFE at concentrations of 0, 12.5, 25, and 50 µg/mL for 24 h. Subsequently, the cells were exposed to 1 μg/mL LPS for an additional 24 h to induce oxidative stress. After the incubation period, the cells were harvested by centrifugation at 1500× *g* for 10 min at 4 °C. The cell pellets were resuspended in cold HEPES buffer (20 mM HEPES, pH 7.2, containing 1 mM EGTA, 210 mM mannitol, and 70 mM sucrose) and homogenized on ice. The homogenates were then centrifuged again at 1500× *g* for 5 min at 4 °C to remove debris, and the resulting supernatants were collected for analysis. For the SOD assay, 10 μL of each sample was mixed with 200 μL of radical detector solution and 20 μL of xanthine oxidase working solution per well in a 96-well plate. The reaction mixtures were incubated at room temperature for 30 min, and absorbance was measured at 450 nm using a microplate reader. The concentration of SOD was calculated based on a standard curve generated from known standards.

### 2.10. GSH Assay

The intracellular glutathione (GSH) content was quantified using the Glutathione Assay Kit (Cayman Chemical, Ann Arbor, MI, USA), based on the spectrophotometric glutathione reductase recycling method described in the manufacturer’s instructions. Cells were seeded at a density of 2 × 10^5^ cells per dish and treated with various concentrations of LFE for 24 h, followed by stimulation with 1 μg/mL LPS for an additional 24 h. Post-treatment, the cells were collected by centrifugation at 1500× *g* for 10 min at 4 °C. The cell pellets were washed twice with cold 50 mM phosphate buffer (pH 7.0, containing 1 mM EDTA), and then lysed by mechanical homogenization. After homogenization, the samples were centrifuged at 10,000× *g* for 15 min at 4 °C. The resulting supernatants were mixed with an equal volume of metaphosphoric acid solution (5 g of metaphosphoric acid in 50 mL of distilled water) and vortexed thoroughly. After standing for 5 min at room temperature, the mixtures were centrifuged again at 3000× *g* for 5 min to remove precipitates. The clarified supernatants were neutralized by adding 50 μL of TEAM reagent (4 M triethanolamine) per mL of sample. For GSH measurement, assay cocktail reagents were added to the samples according to the manufacturer’s guidelines, and the reaction was incubated at room temperature for 25 min. Absorbance was read between 405 and 414 nm using a microplate reader, and GSH concentrations were calculated from a standard calibration curve.

### 2.11. Total RNA Extraction and qPCR

Total RNA was isolated from differentiated SH-SY5Y cells using the Total RNA Purification Kit (GeneMark, Taichung, Taiwan) according to the manufacturer’s protocol. Cells were pre-treated with various concentrations of LFE for 18 h prior to RNA extraction. Subsequently, 20 ng of total RNA was reverse transcribed into complementary DNA (cDNA) using the iScript™ cDNA Synthesis Kit (Bio-Rad, Hercules, CA, USA). Quantitative real-time PCR was performed to evaluate the expression levels of genes associated with melatonin synthesis, including tryptophan hydroxylase-1 (TPH1), dopa decarboxylase (DDC), aralkylamine N-acetyltransferase (AANAT), and acetylserotonin O-methyltransferase (ASMT). GAPDH served as the internal reference gene for normalization. The relative gene expression levels were calculated using the 2^−ΔΔCT^ method [29]. The specific primer sequences used in the experiments are listed in Table 1.

### 2.12. Serotonin Measurement

The serotonin concentrations in cells were determined using a commercially available ELISA kit (Elabscience Biotechnology, Wuhan, Hubei, China), following the manufacturer’s instructions. QGP-1 cells, which serve as a validated model for human enterochromaffin (EC) cells and represent an excellent system for studying serotonin (5-HT) secretion mechanisms, were selected as the established in vitro model for evaluating serotonin synthesis [30,31]. The cells were seeded at a density of 1 × 10^6^ cells per dish and treated with different concentrations of LFE for 24 h. After treatment, the cells were harvested by trypsin-EDTA digestion, collected by centrifugation at 1000× *g*, and washed three times with PBS to remove residual media and reagents. The cells were then subjected to three cycles of freezing and thawing to ensure complete lysis. Following cell disruption, the samples were centrifuged at 1500× *g* for 10 min at 4 °C to obtain the supernatants. The collected supernatants were analyzed for serotonin content using the ELISA kit. Absorbance was measured at 450 nm with a microplate reader, and serotonin concentrations were calculated based on the standard curve generated with known standards.

### 2.13. Statistical Analysis

All data are presented as mean ± SEM deviation from two to three independent experiments. Statistical analysis was performed using GraphPad Prism (Version 10.4.1). ANOVA with Dunnett’s test was used for multiple-group comparisons. Statistical significance was defined as *p* < 0.05.

## 3. Results

### 3.1. Bioactive Compounds in LFE

To better understand the potential health benefits of LFE, we first analyzed its phytochemical composition, focusing on key bioactive compounds with known antioxidant and anti-inflammatory properties. Component analysis (Table 2) and HPLC chromatograms (Figure 1) showed that LFE contained various bioactive compounds, including phenolic compounds and corilagin.

### 3.2. Reduction in Inflammatory Markers

To ensure the extract’s safety for subsequent experiments, we first conducted cell viability assays to determine non-toxic concentrations. As shown in Figure 2A, the cell viability assays showed that LFE concentrations below 50 μg/mL had no significant cytotoxicity. Based on these results, subsequent experiments used concentrations of 12.5, 25, and 50 μg/mL.

As our primary investigation of LFE’s biological activity, we first examined its anti-inflammatory effects by measuring pro-inflammatory cytokine production in RAW264.7 macrophages and BV-2 microglial cells stimulated with LPS, a potent inflammatory trigger. The LPS stimulation of RAW264.7 cells significantly increased TNF-α levels from 2058 ± 345 pg/mL to 6498 ± 646 pg/mL. Treatment with medium and high doses of LFE reduced these levels to 4696 ± 207 pg/mL and 3010 ± 694 pg/mL, respectively (Figure 2B). In BV-2 cells, even low doses of LFE effectively decreased LPS-induced TNF-α secretion (Figure 2C).

LPS stimulation elevated IL-6 levels from 110 ± 19 pg/mL to 673 ± 59 pg/mL in RAW264.7 cells. LFE treatment significantly reduced these levels in a dose-dependent manner to 378 ± 27 pg/mL (low dose), 172 ± 34 pg/mL (medium dose), and 115 ± 21 pg/mL (high dose) (Figure 2D). Similar IL-6 reduction was observed in BV-2 cells (Figure 2E). In supplementary experiments, individual treatment with corilagin or ellagic acid under LPS stimulation significantly reduced IL-6 production in RAW 264.7 cells, while both TNF-α and IL-6 were significantly suppressed in BV-2 cells (see Appendix A), indicating that these compounds contribute to the anti-inflammatory effects of LFE.

These findings demonstrate that LFE has strong anti-inflammatory properties by suppressing pro-inflammatory cytokine production in both macrophages and microglial cells.

### 3.3. Reduction in Nitric Oxide (NO) Expression

To further assess the anti-inflammatory potential of LFE, we examined its effect on nitric oxide (NO) production in immune cell models stimulated with LPS. In our experiments with RAW264.7 macrophages and BV-2 microglial cells, LPS stimulation significantly increased NO production, mimicking the inflammatory state. Treatment with LFE at low, medium, and high concentrations effectively suppressed NO levels in both cell types (Figure 3), suggesting that LFE may contribute to attenuating the inflammatory response by modulating inducible nitric oxide synthase (iNOS) expression and reducing oxidative stress, even at low doses. Similar to LFE treatment, corilagin or ellagic acid significantly reduced NO production under LPS stimulation in both RAW 264.7 and BV-2 cell models (see Appendix A).

### 3.4. Reduction in Reactive Oxygen Species (ROS) and Enhancement of Antioxidant Enzymes

To evaluate the antioxidant potential of LFE, intracellular ROS levels were measured in LPS-stimulated RAW 264.7 and BV2 cells following LFE treatment. Fluorescence intensity served as an indicator of the oxidative stress levels in the cells.

Our results showed that LPS stimulation significantly increased the intracellular ROS levels in both RAW264.7 and BV-2 cells. Treatment with LFE effectively suppressed this ROS production in a concentration-dependent manner (Figure 4A,B), demonstrating its strong antioxidant capacity. In addition, Appendix A showed that corilagin treatment significantly decreased ROS production in both RAW 264.7 and BV-2 cells under LPS stimulation, while ellagic acid showed a significant reduction only at the higher concentration (25 μM), consistent with the effects observed with LFE.

To understand the molecular mechanisms behind this antioxidant effect, we examined key antioxidant enzymes. LFE treatment enhanced the expression of GPx, particularly in RAW264.7 cells, though the effect was more modest in BV-2 cells (Figure 4C,D). Furthermore, LFE treatment enhanced both SOD activity (Figure 4E,F) and intracellular glutathione (GSH) levels (Figure 4G,H). These findings suggest that LFE strengthens cellular antioxidant defenses through multiple pathways.

### 3.5. Enhancement of Melatonin-Synthesis-Related Enzymes

To explore the potential regulatory effects of LFE on melatonin biosynthesis, we measured the expression levels of four key enzymes, including TPH1, DDC, AANAT, and ASMT, in LPS-stimulated cells following LFE treatment. Our results demonstrated that LFE treatment significantly enhanced the expression of all key enzymes: tryptophan hydroxylase 1 (TPH1, 2.34-fold, SD = 0.61), DOPA decarboxylase (DDC, 2.12-fold, SD = 0.21), aralkylamine N-acetyltransferase (AANAT, 2.42-fold, SD = 0.01), and acetylserotonin O-methyltransferase (ASMT, 4.25-fold, SD = 0.67) compared to controls (Figure 5A–D). Furthermore, serotonin secretion increased significantly (by approximately three- to four-fold) in the extract-treated groups (Figure 5E). This suggests that LFE may enhance both serotonin and melatonin production, potentially benefiting sleep quality.

## 4. Discussion

In this study, we demonstrated that LFE exerts multiple bioactive effects in vitro, including anti-inflammatory, antioxidant, and neuromodulatory activities. Our phytochemical analysis revealed that LFE is rich in polyphenols, flavonoids, and specific compounds such as gallic acid, corilagin, and ellagic acid, which are known for their diverse bioactivities. These findings support the potential application of LFE as a natural bioactive agent for managing various health conditions associated with oxidative stress and inflammation.

Edible vegetables and fruits contain diverse bioactive compounds that can help to prevent various stress-induced diseases. OPCs have been reported to exhibit a stronger antioxidant capacity than vitamin C and E [32]. Phenolic compounds from plants, such as phenolic acids and flavonoids, have been confirmed to possess antioxidant and radical scavenging activities [33]. Additionally, corilagin and gallic acid found in medicinal plants have demonstrated anti-depressive and anxiolytic effects [34].

Compared with previous studies focusing primarily on longan fruit pulp or seed-derived polysaccharides [17,18,22], our work uniquely characterizes the underexplored flower component. Notably, the presence of corilagin, a polyphenol with well-documented anti-inflammatory and neuroprotective properties, adds to the scientific value of our extract. Corilagin has been shown to inhibit pro-inflammatory cytokines and modulate oxidative stress in models of liver injury and neurodegeneration [35,36], suggesting that its presence in LFE could contribute to the observed cellular effects.

Nitric oxide (NO), produced by iNOS, is crucial in inflammatory responses and immune regulation [37]. During inflammation, pro-inflammatory cytokines and bacterial endotoxins trigger iNOS expression in immune cells, leading to sustained NO production. While normal NO levels are essential for cell signaling, excess NO creates reactive nitrogen species that damage cellular components [38]. This oxidative stress contributes to the chronic inflammation seen in conditions like atherosclerosis, asthma, and neurodegenerative diseases [39,40,41].

The anti-inflammatory effects of LFE were evidenced by a dose-dependent reduction in TNF-α, IL-6, and nitric oxide production in LPS-stimulated macrophages and microglia. These immune cell models are relevant to both peripheral and central inflammation. While melatonin pathway modulation was a key focus, our data suggest that the extract’s broader immunoregulatory effects may extend beyond neuroinflammatory contexts. This highlights its potential applicability in chronic inflammatory conditions such as metabolic syndrome, cardiovascular diseases, and aging-related immune dysregulation.

Additionally, LFE enhanced the activity of major antioxidant enzymes, including GPx and SOD, and elevated intracellular glutathione levels. These outcomes align with previous findings on phenolic-rich plant extracts that protect cells against oxidative stress via both direct radical scavenging and the upregulation of endogenous defense systems [33,42]. Such dual action is particularly advantageous for addressing diseases where oxidative stress is a central pathological driver, such as atherosclerosis and diabetes.

Importantly, we demonstrated that LFE upregulated the expression of four key enzymes involved in melatonin biosynthesis, including TPH1, DDC, AANAT, and ASMT, as well as serotonin secretion [43]. These findings reinforce the potential of LFE in supporting sleep health. However, considering melatonin’s broader physiological roles—including immune modulation, mitochondrial function, and circadian regulation—this pathway may underlie a wide array of health effects. Future studies should explore whether the observed effects translate to systemic outcomes in animal or clinical models. In addition, in silico analyses, such as molecular docking or computational target prediction, could be used to further elucidate the interactions between LFE compounds and inflammatory or neuromodulatory pathways.

In addition to identifying LFE’s bioactive components, we conducted the fingerprint profiling of samples sourced from various geographic regions, including Taiwan, Thailand, and China. Across all tested batches, key compounds—total phenolics, total flavonoids, OPCs, and corilagin—were consistently detected, as summarized in Table 2. A novel analytical assay was developed to support quality control and purity verification, and a patent application for this technique is currently in progress.

While our results are promising, this study has several limitations. First, all assays were conducted in vitro, and the in vivo pharmacokinetics or bioavailability of LFE components remain unknown. Second, we did not evaluate potential synergistic or antagonistic interactions among the identified phytochemicals. Finally, the mechanisms driving enzyme gene upregulation remain to be clarified.

In conclusion, this study provides novel evidence for the multifaceted bioactivity of longan flower extract, supporting its development as a natural agent for managing oxidative stress, inflammation, and possibly sleep-related health issues. By focusing on a traditionally underutilized plant part, our work contributes to the broader search for sustainable, plant-based therapeutics and highlights the need for further investigation into LFE’s mechanisms and clinical applicability.

## Figures and Tables

**Figure 1 biomedicines-13-01588-f001:**
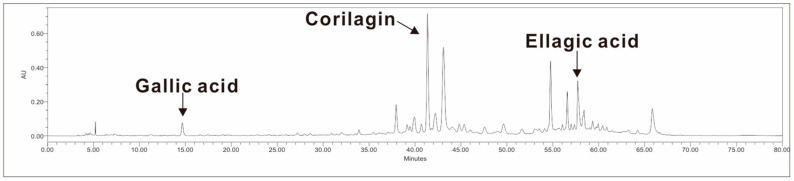
Representative HPLC chromatogram of LFE. Representative HPLC chromatogram showing the major bioactive compounds identified in LFE. The analysis revealed the presence of several key compounds, including gallic acid (retention time ~15 min), corilagin (retention time ~41 min), and ellagic acid (retention time ~58 min).

**Figure 2 biomedicines-13-01588-f002:**
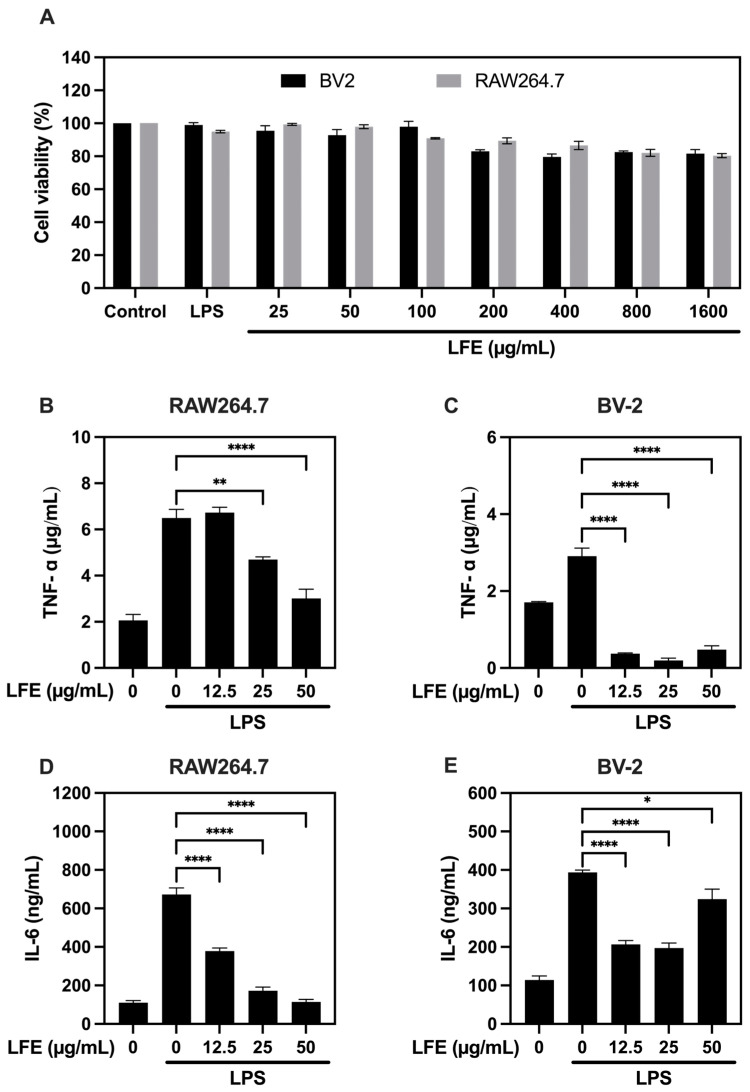
Anti-inflammatory effects of LFE in LPS-stimulated cells. (**A**) Cell viability of BV2 and RAW264.7 cells exposed to different LFE concentrations; TNF-α production in RAW264.7 cells (**B**) and BV-2 (**C**) treated with LPS and various LFE concentrations; IL-6 production in RAW264.7 cells (**D**) and BV-2 (**E**) with LPS and LFE treatment. The quantification data are presented as mean values (±SEM) derived from experiments performed 3 independent times under identical conditions. * *p* < 0.05, ** *p* < 0.01, **** *p* < 0.0001 (one-way ANOVA with Dunnett’s multiple comparisons test). LFE, longan flower extract; LPS, lipopolysaccharide.

**Figure 3 biomedicines-13-01588-f003:**
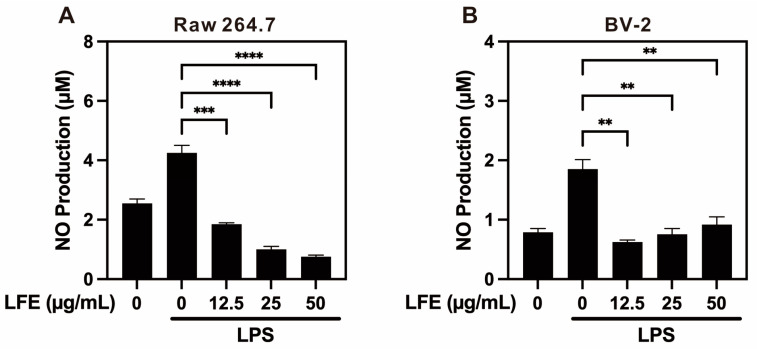
Suppression of nitric oxide production by longan flower extract in immune cells. The effect of LFE on nitric oxide (NO) production was evaluated in inflammatory cell models. (**A**) NO production in RAW264.7 macrophages treated with LPS and varying concentrations of LFE (0–50 μg/mL). (**B**) NO production in BV-2 microglial cells under similar treatment conditions. Data are presented as mean values ± SEM (n = 2). ** *p* < 0.01, *** *p* < 0.001, **** *p* < 0.0001 (one-way ANOVA with Dunnett’s multiple comparisons test). LFE, longan flower extract; LPS, lipopolysaccharide.

**Figure 4 biomedicines-13-01588-f004:**
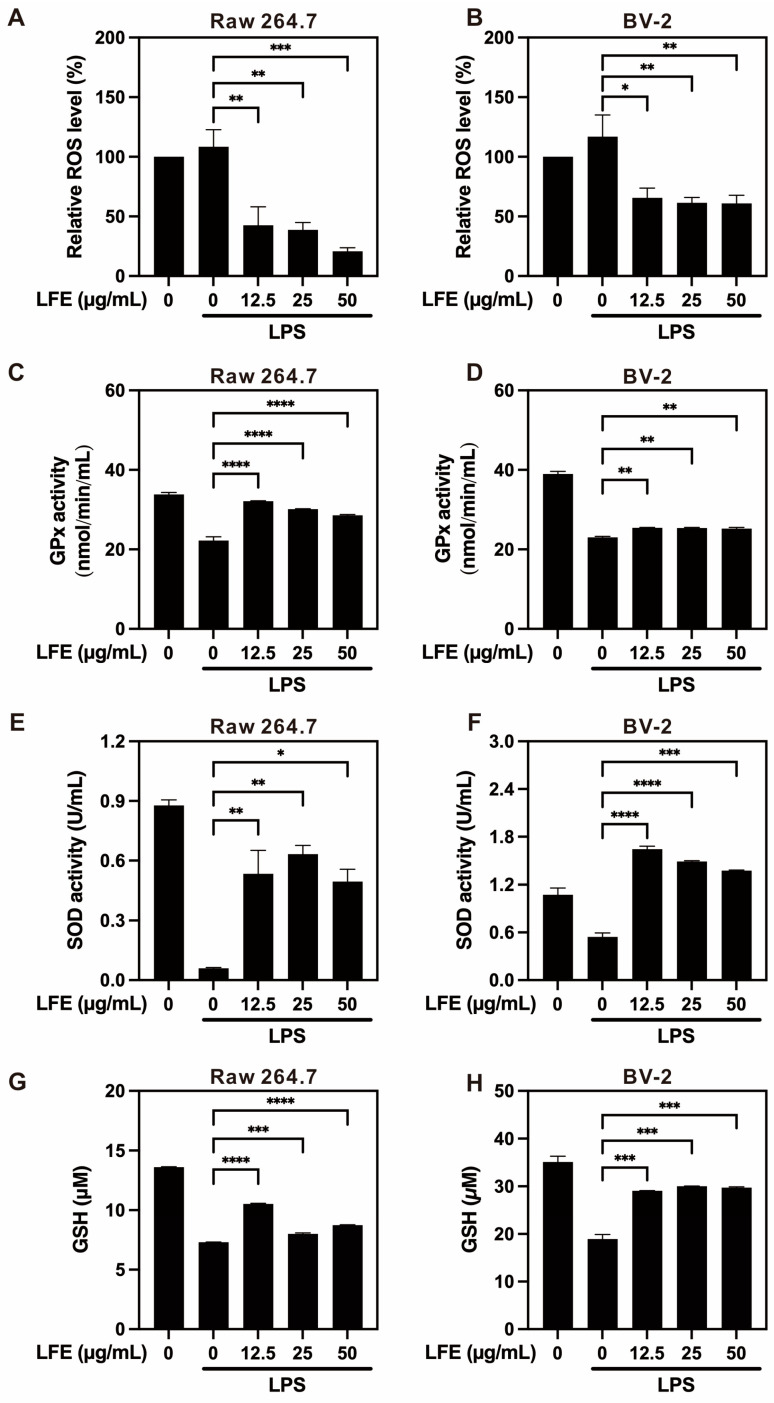
Longan flower extract reduces oxidative stress and enhances antioxidant enzyme activity in immune cells. LFE’s effects on oxidative stress markers and antioxidant enzyme activities were evaluated in RAW264.7 macrophages and BV-2 microglial cells. The figure shows: (**A**,**B**) ROS levels, (**C**,**D**) GPx expression, (**E**,**F**) SOD activity, and (**G**,**H**) GSH levels in RAW264.7 and BV-2 cells after LPS stimulation with different concentrations of LFE treatment. The quantification data are presented as mean values (±SEM) derived from experiments performed 3 independent times (**A**–**D**) and 2 independent times (**E**–**H**) under identical conditions. * *p* < 0.05, ** *p* < 0.01, *** *p* < 0.001, and **** *p* < 0.0001 (one-way ANOVA with Dunnett’s multiple comparisons test).

**Figure 5 biomedicines-13-01588-f005:**
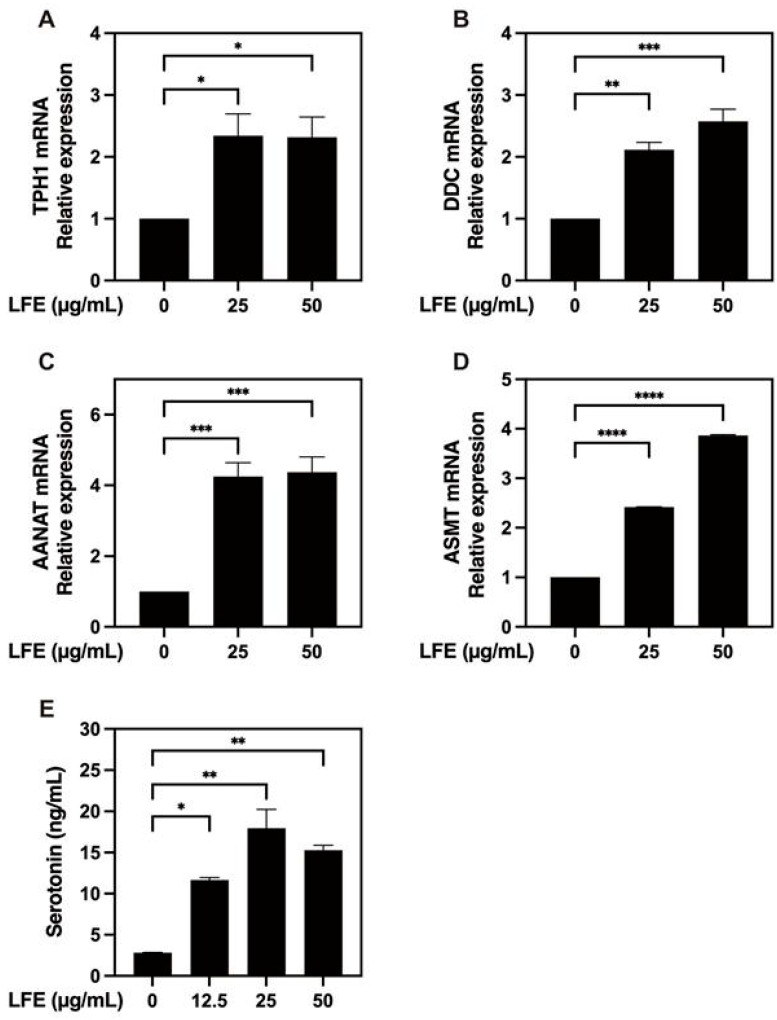
Enhancement of melatonin-synthesis-related enzyme expression by LFE treatment. The expression levels of key enzymes involved in melatonin biosynthesis were analyzed following LFE treatment. The panels show relative mRNA expression of: (**A**) tryptophan hydroxylase 1 (TPH1), (**B**) DOPA decarboxylase (DDC), (**C**) aralkylamine N-acetyltransferase (AANAT), and (**D**) acetylserotonin O-methyltransferase (ASMT). The quantitative results are presented as relative mRNA levels compared to control groups (mean ± SEM, n = 3). (**E**) QGP-1 cells were treated with different concentrations of LFE (0, 12.5, 25, and 50 μg/mL) and serotonin levels were quantified (mean ± SEM, n = 2). * *p* < 0.05, ** *p* < 0.01, *** *p* < 0.001, and **** *p* < 0.0001 (one-way ANOVA with Dunnett’s multiple comparisons test).

**Table 1 biomedicines-13-01588-t001:** Primer sequences for *TPH1*, *DDC*, *AANAT*, and *ASMT*.

Gene	Primer Forward Sequence	Primer Reverse Sequence
*GAPDH*	CTGGGCTACACTGAGCACC	AAGTGGTCGTTGAGGGCAATG
*TPH1*	ACGTCGAAAGTATTTTGCGGA	ACGGTTCCCCAGGTCTTAATC
*DDC*	ACCACAACATGCTGCTCCTTT	ATCAACGTGCAGCCATATGTCT
*AANAT*	TGTCAGCGCCTTTGAGATCG	CTCTGGACATAGGGTCAGGAA
*ASMT*	GGGCAGACGGAAAGTGCTC	CTGCACAAGCATGTTCAGAGA

**Table 2 biomedicines-13-01588-t002:** Chemical composition of longan flower ethanol extract.

Test Items (Unit)	Value
Total phenolic content (mg/g)	171.4 ± 5.9
Total flavonoid (mg/g)	233.3 ± 16.7
Oligomeric proanthocyanidins (OPCs) (mg/g)	15.9 ± 2.1
Corilagin (mg/g)	37.7 ± 4.8

## Data Availability

Pre-processed raw data are available and can be shared with interested parties upon reasonable request. For access, please contact the corresponding author Yu-Wei Chang (email: golden3p@csmu.edu.tw).

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
