# Peer review of "Longan Flower Ethanol Extract, Dimocarpus longan Lour, Mitigates Oxidative Damage and Inflammatory Responses While Promoting Sleep-Related Enzymes in Cell Models"

_biomedicines, 2025, doi:10.3390/biomedicines13071588_

Round 1
Reviewer 1 Report
Comments and Suggestions for Authors
Please refer to the manuscript

can be improved
Author Response
Thank you for your helpful comments on our current work. We have revised the manuscript and provided our responses in the following section.
Comment 1: Revise the title of the study.
Response: We have revised the title as “Longan Flower Ethanol Extract, Dimocarpus Longan Lour, Mitigates Oxidative Damage and Inflammatory Responses While Promoting Sleep-Related Enzymes in Cell Models”. (Please see Line 2)
Comment 2: The sentence “These findings suggest LFE's significant potential as a natural therapeutic supplement for addressing modern lifestyle-related health challenges.” – especially?
Response:
Thank you for your insightful comment. We agree that the original sentence was too general. To clarify, we have revised it to specify the key health challenges targeted by LFE. The updated sentence now reads:
“These findings suggest that LFE holds significant potential as a natural therapeutic supplement, particularly for alleviating sleep disturbances, reducing oxidative stress, and modulating inflammatory responses associated with modern lifestyle-related health conditions.” (Please see Line 30-33)
Comment 3:
The sentence “This physiological deterioration, coupled with age-related increases in life stressors, creates a negative feedback loop whereby diminished sleep quality adversely affects various aspects of health, including cognitive function, metabolic homeostasis, and immune system regulation.” – ?
Response:
Thank you for pointing this out. We agree that the term “negative feedback loop” was inaccurately used in this context. To improve clarity, we have revised the sentence to better describe the compounding effects between sleep quality and health deterioration. The updated version reads:
“This physiological decline, combined with age-related life stressors, creates a vicious cycle where poor sleep disrupts cognitive function, metabolism, and immune response, which in turn leads to even worse sleep quality.” (Please see Line 46-49)
Comment 4:
The sentence “Animal studies have shown that sleep deprivation increases oxidative stress and causes tissue inflammation.” – tissue?
Response:
Thank you for your suggestion. We agree that the term “tissue inflammation” lacked specificity. To improve clarity and scientific accuracy, we revised the sentence to:
“Animal studies have shown that sleep deprivation increases oxidative stress and induces systemic and neuroinflammatory responses.” (Please see Line 53-54)
Comment 5:
The sentence “These concerns highlight the growing need for non-pharmacological approaches to address modern lifestyle-related health challenges.” – particularly?
Response:
Thank you for your valuable feedback. We agree that the original statement was too general. To clarify the types of health challenges we are addressing, we revised the sentence as follows:
“These concerns highlight the growing need for non-pharmacological approaches to address modern lifestyle-related health challenges, particularly those involving sleep disturbances, chronic inflammation, and oxidative stress.” (Please see Line 57-59)
Comment 6:
The phrase “have well-documented anti-inflammatory and antioxidant properties that help reduce metabolic disease risks.” – re-write to correct English.
Response:
Thank you for the constructive feedback. We have rewritten the entire sentence for improved clarity and flow while maintaining the original meaning. The revised sentence now reads:
“Active compounds found in fruits, vegetables, and edible flowers, such as phenolic acids, flavonoids, and anthocyanins, are widely recognized for their anti-inflammatory and antioxidant effects, which help lower the risk of developing metabolic diseases.” (Please see Line 61-64)
Comment 7:
Suggest revising “longan pulp” to “longan fruit pulp” for consistency with the cited references.
Response:
Thank you for your thoughtful suggestion. We have updated the terminology from “longan pulp” to “longan fruit pulp” throughout the relevant section to improve clarity. (Please see Line 70 and 72)
Comment 8:
Suggest adding a citation to support the statement: “Although some traditional practices include brewing longan flowers as herbal tea...”
Response:
Thank you for your suggestion. As we were unable to locate a reliable reference to support this traditional use, we have removed the sentence regarding the herbal tea practice to maintain scientific accuracy and clarity in the revised manuscript. (Please see Line 73-77)
Comment 9:
(1) What about its side effects and toxicology?
(2) The phrase “preventing lifestyle-related diseases” is too broad—please make it specific.
Response:
Thank you for these valuable suggestions. We agree that the original sentence was too broad and did not acknowledge the need for safety evaluation. We have revised the text to specify the targeted health concerns—namely, oxidative stress, inflammation, and sleep-related disturbances—and included a statement recognizing that toxicological assessment remains an important area for future investigation. The revised sentence now reads:
“Our goal is to evaluate whether longan flower extract could be developed into a functional nutraceutical to promote health, particularly by alleviating oxidative stress, modulating inflammation, and enhancing sleep-related biochemical pathways. While this study focuses on in vitro efficacy, future investigations will be needed to assess the safety profile and potential toxicological effects of LFE in vivo.” (Please see Line 83-87)
Comment 10:
Need more details: flower collection location, time, condition, maturity stage, etc. Also, specify the extraction solvent ratio and equipment used for heating.
Response:
Thank you for the helpful suggestions. We have revised the “Preparation of Longan Flower Extract” section to include details on the flower collection site, timing, maturity stage, extraction solvent concentration and ratio, and the equipment used for heating and concentration. The updated paragraph now reads:
“Longan (Dimocarpus longan Lour.) flowers were collected in Taichung, Taiwan, during full bloom in April 2023 from pesticide-free farms. Only mature, fully opened flowers were used. The fresh flowers were air-dried at 50°C in the shade for 24 hours and then pulverized into coarse powder. The dried flower powder was extracted with ethanol at a material-to-solvent ratio of 1:15 (w/v). The mixture was heated at 65°C for 60 minutes using a thermostatic water bath. The resulting extract was filtered through a 400-mesh filter, concentrated under reduced pressure using a rotary evaporator, and finally spray-dried to obtain powdered longan flower extract (LFE).” (Please see Line 90-99)
Comment 11:
(1) Is the sample used in the polyphenol/flavonoid/OPC assays the spray-dried powder?
(2) What was the reason for the modified flavonoid quantification method?
(3) Why didn’t you use the spray-dried powder for HPLC, as mentioned earlier?
(4) Table 1 is not necessary or informative in its current form.
Response:
Thank you for these insightful comments. We have revised the Methods section to clearly state that the polyphenol, flavonoid, and OPC assays were conducted using reconstituted spray-dried longan flower extract (LFE) powder. The spray-dried form was chosen for these assays to ensure consistency with the product intended for nutraceutical development.
The flavonoid quantification method was slightly modified from the original procedure to improve the sensitivity and reproducibility when working with diluted plant extract matrices and to minimize interference from polysaccharides and other non-phenolic compounds present in LFE.
Regarding the HPLC analysis, we acknowledge the apparent inconsistency. The extract for HPLC profiling was intentionally prepared from the original air-dried longan flowers rather than the spray-dried powder. This decision was based on preliminary validation work, which indicated that some thermolabile or volatile compounds might degrade or be lost during spray-drying, potentially affecting the accuracy of component quantification. Using the original dried flower material ensures a more representative profile of naturally occurring polyphenols. We have clarified this rationale and revised the corresponding section in the manuscript accordingly.
Finally, we agree that Table 1, which detailed the HPLC gradient elution program, provided limited added value to readers and may distract from the core data. As such, we have removed Table 1 from the revised manuscript to streamline the Methods section and maintain focus on relevant experimental information. (Please see Line 100-131)
Comment 12:
Please acknowledge the source of the BV-2 mouse microglial cells.
Response:
Thank you for the suggestion. Based on your comment, we have added an acknowledgment to credit Taichung Veterans General Hospital for their generous providing of the BV-2 cell line. (Please see Line 461-462)
Comment 13:
Please specify the concentrations used for LFE and mention positive and negative controls in the TNF-α and IL-6 assay.
Response:
Thank you for your helpful suggestion. We have revised Section 2.5 to include the specific concentrations of LFE used in the experiment (12.5, 25, and 50 µg/mL). We also clarified that untreated cells served as the negative control, and LPS-treated cells (without LFE) served as the positive control. These details have been incorporated to enhance clarity and reproducibility. (Please see Line 142-144)
Comment 14:
The first paragraph of Section 3.1 is more appropriate for the Discussion section than the Results.
Response:
Thank you for your insightful suggestion. We have revised Section 3.1 to begin directly with the experimental findings and relocated the interpretive, literature-based paragraph to the beginning of the Discussion section, where it better supports the interpretation of our results. (Please see Line 271-273)
Comment 15:
In general, the total phenolic content (TPC) is expected to be higher than the total flavonoid content (TF), yet your results in Table 3 show the opposite.
Response:
Thank you for your insightful observation regarding the relative levels of total phenolic content (TPC) and total flavonoid content (TF) in our LFE samples. We acknowledge that in many plant extracts, TPC is typically reported as higher than TF. However, in our study, repeated analyses across multiple LFE batches consistently showed higher TF than TPC levels.
This result may be attributed to the specific phytochemical composition of longan flowers, which appear to be particularly rich in flavonoids relative to non-flavonoid phenolics. Additionally, the differences in assay methods and their respective reactivities could also contribute to this outcome. While the Folin–Ciocalteu method used for TPC detection broadly reacts with all phenolic hydroxyl groups, the aluminum chloride colorimetric method for TF is more selective toward flavonoid structures. Thus, the elevated TF in our data may reflect a true abundance of flavonoids in the extract, consistent with the flower’s phytochemical profile.
We have carefully verified these measurements across multiple preparations and believe the findings accurately reflect the unique composition of our LFE material.
Comment 16:
Please revise the term “concentrations” to “doses” in Lines 266–267.
Response:
Thank you for the suggestion. We have revised the term “concentrations” as “doses” in Lines 266–267 to improve precision and consistency.
Comment 17:
Clarify the grouping and labeling in Figure 1, particularly the use of “LPS” across the x-axis.
Response:
Thank you for your helpful comment. We would like to clarify that the figure layout reflects a dose-dependent co-treatment design. Specifically, all bars to the right of the initial control represent groups treated with LPS (1 µg/mL), including the LPS-only group and the three subsequent groups co-treated with LFE at 12.5, 25, and 50 µg/mL, respectively. The label “LPS” below the axis indicates that these four groups were all exposed to LPS. The first bar (LFE = 0 µg/mL without LPS) serves as the untreated control. (Please see new Figure 2)
Comment 18:
In Section 3.3, the first paragraph should be moved to the Discussion as it is interpretive. Additionally, the sentence in Line 298 should be revised for grammar and clarity, and the figure legend for Figure 2 should abbreviate “Longan Flower Extract” as “LFE.”
Response:
Thank you for your constructive feedback. According to your suggestion, we have integrated the first paragraph of Section 3.3 to the Discussion section, as it offers interpretation rather than direct results. We have also revised the sentence in Line 298 for clarity and grammatical correctness. (Please see Line 316-326)
The new version reads:
“These results suggest that LFE may contribute to attenuating the inflammatory response by modulating iNOS expression and reducing oxidative stress, even at low doses.”
Furthermore, we have updated the legend for Figure 2 to abbreviate “Longan Flower Extract” as “LFE” to ensure consistency throughout the manuscript.
Comment 19:
In Line 334, please remove repeated abbreviation definitions (e.g., LFE, LPS) if they are already clearly defined in the figure legends.
Response:
Thank you for your suggestion. We agree that the repeated abbreviation definitions in the main text are unnecessary when already included in the figure legends. We have reviewed the manuscript and removed redundant abbreviation explanations (e.g., for LFE and LPS) to enhance readability and avoid repetition.
Comment 20:
In Figure 4, only the 25 and 50 µg/mL doses of LFE were tested for melatonin-related enzyme gene expression (TPH1, DDC, AANAT, ASMT). The 12.5 µg/mL dose appears to be missing.
Response:
Thank you for your observation. In this experiment, we focused specifically on the 25 and 50 µg/mL doses of LFE because these concentrations demonstrated more pronounced biological activity in earlier assays. Our aim was to evaluate the gene expression of melatonin-related enzymes at doses where functional effects were most evident. We believe this focused approach allowed for clearer interpretation of the dose-dependent effects of LFE on melatonin biosynthesis.
Comment 21:
The current Discussion section reads like a summary or conclusion. It would benefit from deeper interpretation of the results, integration with existing literature, and broader discussion of the biological implications beyond melatonin and neuroinflammation.
Response:
Thank you for your constructive suggestion. In the revised manuscript, we have thoroughly enhanced the Discussion section to strengthen its scientific depth and relevance. The new version goes beyond summary by providing detailed interpretation of our findings within the context of existing literature. To extend the research value of our work, we have also emphasized future directions in the revised version, such as in silico analyses, which will strengthen the evidence for LFE's biological effects. (Please see revised Discussion Section)

Reviewer 2 Report
Comments and Suggestions for Authors
The present study investigated Dimocarpus longan Lour. flower extract (LFE), a plant traditionally used in East Asian medicine. Results showed that LFE enhances melatonin synthesis pathways, reduces key inflammatory markers (TNF-α, IL-6, and NO), and strengthens antioxidant defenses against reactive oxygen species (ROS). These results suggest that LFE has significant potential as a natural therapeutic supplement for addressing lifestyle-related health challenges. This well-conducted study provides new and important information about the use of this plant species. Therefore, I suggest accepting it for publication in Biomedicines after making the following minor revisions:
1. The reason for using ethanol to prepare the extract is unclear. Is this solvent used in traditional medicine to prepare extracts?
2. Table 1 must be withdrawn, and the data can be briefly described in the text.
3. The chemical composition of the studied extract (LFE) is incomplete. Several chromatographic bands appear in the HPLC chromatogram and must be identified. Use MS techniques in association with HPLC for this procedure.
4. Since coliragin, ellagic acid, and gallic acid were commercially available, why were the effects of these compounds not evaluated individually?
Author Response
Thank you for your helpful comments on our current work. We have revised the manuscript and provided our responses in the following section.
Comment 1: The reason for using ethanol to prepare the extract is unclear. Is this solvent used in traditional medicine to prepare extracts?
Response:
Thank you for your insightful comment. We have clarified in the revised manuscript that ethanol was chosen as the extraction solvent due to its dual relevance in both traditional and modern phytopharmaceutical contexts. Ethanol is commonly used in traditional herbal preparations and is well-documented in scientific literature for its efficiency in extracting polyphenolic compounds. We have added this explanation to the Methods section to improve transparency regarding our extraction rationale.
Comment 2: Table 1 must be withdrawn, and the data can be briefly described in the text.
Response:
Thank you for your insightful comment. We agree that Table 1, which detailed the HPLC gradient elution program, provided limited value to readers and could distract from the core data. Therefore, we have removed Table 1 from the revised manuscript to streamline the Methods section and maintain focus on the essential experimental information.
Comment 3:
The chemical composition of the studied extract (LFE) is incomplete. Several chromatographic bands appear in the HPLC chromatogram and must be identified. Use MS techniques in association with HPLC for this procedure.
Response:
Thank you for this important suggestion. We agree that comprehensive chemical profiling is critical for the standardization and quality control of botanical extracts. In our study, we performed fingerprint profiling of LFE samples collected from various geographic sources, including Taichung, Tainan, and Nantou (Taiwan), as well as Thailand and China. Across all sources, the four core bioactive components consistently detected were total phenolics, total flavonoids, oligomeric proanthocyanidins (OPCs), and corilagin.
To further validate compound identity and detect potential impurities, we performed additional MS-coupled chromatographic analysis. The fingerprint data confirmed chemical consistency across samples, although minor uncharacterized peaks suggest possible trace contaminants. However, due to an ongoing patent application involving our proprietary purification and compound authentication process, we are currently limited in the detailed disclosure of full MS results.
Therefore, we have provided a representative chemical profile in the manuscript to demonstrate the major composition. We plan to publish a full analytical characterization report, including MS data, upon completion of the intellectual property process. We appreciate your understanding and believe the current data sufficiently supports the reproducibility and core composition of LFE.
Comment 4:
Since coliragin, ellagic acid, and gallic acid were commercially available, why were the effects of these compounds not evaluated individually?
Response:
Thank you for your valuable comment. In response, we have performed additional experiments to evaluate the individual effects of corilagin and ellagic acid on inflammatory responses. The results, now provided in the Supplementary Figures (sFig.1–sFig.4), demonstrate that both compounds significantly reduced LPS-induced production of pro-inflammatory cytokines, nitric oxide (NO), and reactive oxygen species (ROS) in RAW 264.7 and BV-2 cell models. These effects were consistent with those observed for the LFE treatment in our main study.
These findings support the hypothesis that corilagin and ellagic acid contribute substantially to the anti-inflammatory and antioxidant activities of LFE. We have referenced the new supplementary data in the revised manuscript and appreciate your recommendation, which strengthened the interpretation of our study.

Reviewer 3 Report
Comments and Suggestions for Authors
Dear Authors,
The paper is well written and informative. However, several corrections are required to improve its scientific quality:
1. Please correct some useless English expressions.
2. The abstract and introduction sections are well written. You need only to clarify whether the statistical data concerning the insomnia frequency in the population concerns the entire world or just one country!!! I think it would be interesting to precise that the reported statistics concern a study conducted in Berlin (Germany).
3.The methodology section should be completed as some information is lacking, such as: firstly, the origin of plant material should be given (with geodesic coordinates), the collection date and conditions should also be specified.
Secondly, the reagent's volumes used for each protocol should be specified, such as the ROS staining buffer, the components of the lysis buffer, the sample, radical detector and xanthine oxidase solution, as well as the samples and LFE soncentrations. Thirdly, the calculation procedure of the secondary metabolite's concentrations should be explained. Indeed, equations should be given for content determination of polyphenols, flavonoids, etc, specifying the preparation protocol of calibration curves, for each measurement. Finally, the reference of all methods should be added, especially the 2−ΔΔCT method.
4. The whole remaining manuscript should be edited and reorganized to separate results from discussion, and to have a common thread between the different ideas. Indeed, the discussion should normally be placed after the presentation and interpretation of the results. Moreover, you have separated the results section and the Discussion section. So please put all discussions in the corresponding section and let only the presentation of your results in the 'Results section' !!!
5. Figure 1 should be better dissociated in two parts. Indeed, the chromatogram belongs to the previous subsection concerning the biocompounds and should be presented separately, after the corresponding subsection.
6. The discussion section should be improved. Indeed, results found in your investigation should be compared with the results of previous studies, notably those concerning the pulp of the same studied plant, when possible.
7. Moreover, a Conclusion section should be added after the discussion section.
Please see imperatively the attached file for more comments and more precision.
Good luck.

Author Response
Thank you for your helpful comments on our current work. We have revised the manuscript and provided our responses in the following section.
Comment 1: It is required to clarify whether these statistical data concern the entire world or just one country !!!
I think it would be interesting to precise that reported statistics concern a study conducted in Berlin (Germany).
Response:
Thank you for your helpful observation. We agree that the geographical context of statistical data should be clearly stated. Accordingly, we have revised the sentence to clarify that the data refer specifically to adults in the United States. The revised sentence now reads:
“Current statistical analyses indicate that approximately 30–50% of adults in the United States experience insomnia symptoms.”
Comment 2:
In section 2.1:
- What about the plant material collection? First, the origin of the longan flowers must be given, with precision of geodesic coordinates, and the date and conditions of their acquisition should also be explained.
- Please provide the used ratio. (ethanol)
Response:
Thank you for this important comment. We have revised the manuscript to provide more specific details regarding the plant material collection. The revised version includes that the longan (Dimocarpus longan Lour.) flowers were collected in Taichung, Taiwan, during full bloom in April 2023 from pesticide-free farms, and that only mature, fully opened flowers were used. We have also specified the ethanol-to-material ratio used for extraction in the revised manuscript; this information can be found in Section 2.1.
Comment 3:
In Section 2.2: What about the calculations of these secondary metabolite concentrations? Equations for content determination should be given for polyphenols, flavonoids, and proanthocyanidins, with clarification on the preparation of calibration curves.
Response:
Thank you for your insightful comment. In the revised manuscript, we have clarified that all secondary metabolite assays were performed using standard curves prepared from well-established reference compounds—tannic acid for total polyphenols, quercetin for flavonoids, and catechin for proanthocyanidins. These standard curves were constructed with correlation coefficients (R²) greater than 0.95, ensuring strong linearity and accuracy of the quantification. This addition, now reflected in Section 2.2, aligns with accepted protocols in phytochemical analysis and supports the reliability of our results without disclosing complex formula details.
Comment 4:
In Section 2.4: The samples different concentrations should be given.
Response:
Thank you for your helpful suggestion. We have revised Section 2.4 to clearly specify the concentrations of LFE tested in the cell viability assay, which were 0, 25, 50, 100, 200, 400, 800, and 1600 µg/mL. We have also indicated that LPS-only and medium-only groups were included as controls.
Comment 5:
The volume used for ROS Staining Buffer should be specified.
Response:
Thank you for pointing this out. We have revised Section 2.7 to specify that 100 μL of ROS Staining Buffer was added to each well during the staining procedure.
Comment 6:
Please revise the term “Post-treatment” as “thereafter” and provide the volume of lysis buffer used.
Response:
Thank you for your valuable suggestions. In the revised manuscript, we have replaced the phrase “post-treatment” with “thereafter” to improve clarity. Additionally, we have specified that 100 μL of cold lysis buffer was used to resuspend the cell pellets.
Comment 7:
Please clarify the concentrations of LFE used and specify the volumes of the radical detector and xanthine oxidase solution in Section 2.9.
Response:
Thank you for your helpful suggestion. We have revised Section 2.9 to specify that the concentrations of LFE used were 0, 12.5, 25, and 50 µg/mL. Additionally, we have clarified that for the SOD activity assay, 10 μL of each sample was mixed with 200 μL of radical detector and 20 μL of xanthine oxidase working solution, following the manufacturer’s instructions.
Comment 8:
Please add the reference citation for the used method (indicate the 2−ΔΔCT method) in Section 2.9.
Response:
Thank you for your comment. We have revised the manuscript to include a reference for the 2−ΔΔCT method used in the gene expression analysis.
Comment 9:
In Section 3.1 and 3.5: The discussion should be placed after the presentation and interpretation of the results. Hence, please reorganize your manuscript, especially the Results section, to have a coherent thread between the ideas. Moreover, you have a separate Discussion section, so please put all discussions in this section and let only the presentation of your results in the Results section.
Response:
Thank you for your valuable suggestion. We have carefully revised the manuscript structure to clearly separate the presentation of results from their interpretation. All interpretive or comparative content has been moved to the dedicated Discussion section, ensuring that the Results section now focuses solely on presenting experimental findings. This adjustment improves the clarity and logical order of the manuscript.
Comment 10:
In Figure 1: It would be better to dissociate Figure 1 into two parts. The chromatogram belongs to the previous section concerning the biocompounds and should be presented separately.
Response:
Thank you for your constructive feedback. In accordance with your suggestion, we have revised the manuscript by separating the original Figure 1 into two individual figures. The HPLC chromatogram has been presented independently as the new Figure 1 to accompany the chemical composition section. The data related to LFE’s biological effects, including cell viability and cytokine suppression, have been relocated to the new Figure 2. This change improves the logical flow and alignment of figures with their respective results sections.
Comment 11:
This section has to be improved, and the results found in the present work should be compared with the results of previous studies, notably those concerning the pulp of the same studied plant, when possible. Moreover, a Conclusion section should be added after this one.
Response:
Thank you for your thoughtful recommendations. In response, we have substantially revised the Discussion section to better summarize our key findings and compare them with results from prior studies, particularly those involving longan fruit pulp and other parts of the plant. We also added more explanation about current and future research directions and acknowledged the limitations of our study. Additionally, a separate and concise Conclusion paragraph has been added at the end of the Discussion to clearly highlight the significance of our findings and their potential applications.
Comment 12:
In Abbreviation Section: Please homogeneise the abreviations presentation in this list.
Response:
Thank you for your reminder. We have revised the content of the abbreviation list.

Reviewer 4 Report
Comments and Suggestions for Authors
Dear author,
Thanks for your report. My comments and suggestions are in the attached file. M&M section needs more citation and also rephrase.
Regards

Author Response
Thank you for your helpful comments on our current work. We have revised the manuscript and provided our responses in the following section.
Comment 1:
There are many studies reported the antioxidant activity and, anti cancer and other effects of LFE. Why you did not cite them here to show the gap for your research. Like: https://doi.org/10.1021/jf801155j, Potential roles of longan flower and seed extracts for anti-cancer, Longan flower extract inhibits the growth of colorectal carcinoma, antiobesity and Hypolipidemic Effects of Polyphenol-Rich Longan (Dimocarpus longans Lour.) and Flower Water Extract in Hypercaloric-Dietary Rats
Response:
Thank you for your valuable suggestion. We agree that prior studies have reported promising bioactivities of longan flower extracts. In response, we have revised the Introduction to cite relevant literature on the antioxidant, anticancer, anti-obesity, and hypolipidemic effects of LFE (such as https://doi.org/10.1021/jf801155j and others). These additions help clarify the current state of research while emphasizing that comprehensive studies integrating both phytochemical profiling and mechanistic exploration of LFE, especially in relation to its neuromodulatory and anti-inflammatory properties.
Comment 2: These aims examined before.
Response:
Thank you for your comment. While certain bioactivities of longan flower extract have been previously explored, most studies have focused on general antioxidant or anticancer properties. Our study is distinct in its integration of detailed phytochemical profiling with mechanistic evaluation of LFE’s effects on oxidative stress, inflammation, and melatonin-related pathways in both immune and neuronal cell models. To clarify this novelty, we have revised the aim statement in the manuscript accordingly.
Comment 3:
In Section 2.1: Add reference with more details of ethanol used in the LFE extraction.
Response:
Thank you for your comment. We have added more information about the LFE extraction in the revised version. Please see the updated content in Section 2.1.
Comment 4:
In Section 2.2: More information about HPLC device please.
Response:
Thank you for your comment. We have added more information about the HPLC device in the revised version. Please see the updated content in Section 2.2.
Comment 5:
In Section 2.4: Add reference (of cell viability assay) please.
Response:
Thank you for your comment. We have added reference about cell viability assay in the revised version. Please see the updated content in Section 2.4.
Comment 6:
In Section 3.1, 3.3, 3.4, and 3.5: This sentence belongs to the discussion, not your results.
Response:
Thank you for your observation. We agree with your assessment and have revised the manuscript accordingly. These sentences have been removed from the Results section and appropriately integrated into the revised Discussion section to ensure a clearer separation between data presentation and interpretation.
Comment 7:
“****p<0.0001 is not a common p.”
Response:
Thank you for pointing this out. The asterisk notation (**** for p < 0.0001) was automatically generated by GraphPad Prism (version 10.4.1) as part of the one-way ANOVA analysis followed by Dunnett’s multiple comparisons test. This notation corresponds to an adjusted p-value of less than 0.0001, which is within the accepted convention for highly significant results in Prism-based analyses. To maintain clarity, we have retained this format but are happy to adjust if the editorial team recommends standardizing the notation.
Comment 8:
Poor discussion as you mentioned some of discussions in the results section. Please separate results and discussion sections or combined them. Now, you reported mixed of both cases.
Response:
Thank you for your valuable feedback. In response, we have thoroughly revised the manuscript structure. We carefully separated descriptive data from interpretive commentary by relocating all discussion points from the Results sections into the Discussion section. The revised manuscript now presents a clearer distinction between experimental findings and their interpretation, ensuring logical flow and scientific clarity in line with academic standards. We appreciate your suggestion, which helped improve the manuscript’s overall structure and readability.
Comment 9:
The number of cited references is low. Use more recent related papers in Introduction, M&M and Discussion.
Response:
Thank you for your valuable suggestion. We have thoroughly reviewed and updated the manuscript by incorporating additional and more recent references, particularly in the Introduction, Materials and Methods, and Discussion sections.

Round 2
Reviewer 4 Report
Comments and Suggestions for Authors
Dear author,
Thanks for the revised manuscript. Now, it is much better.
Please check Line 343: sFig.4 showed...
Regards
Author Response
Comment 1: Please check Line 343: sFig.4 showed...
Response: Thank you for your insightful observation. We have carefully reviewed the data presented in sFig.4. As suggested, we modified the manuscript to reflect that ellagic acid demonstrated a suppressive effect on ROS production only at higher concentrations, whereas corilagin showed a significant reduction across both cell models. The revised statement now accurately represents the experimental results. (Please see Line 342-345)